# Blocking transport resonances via Kondo many-body entanglement in quantum dots

Michael Niklas[1], Sergey Smirnov[1], Davide Mantelli[1], Magdalena Margańska[1], Ngoc-Viet Nguyen[2], Wolfgang Wernsdorfer[2], Jean-Pierre Cleuziou[2,3] & Milena Grifoni[1]

Many-body entanglement is at the heart of the Kondo effect, which has its hallmark in quantum dots as a zero-bias conductance peak at low temperatures. It signals the emergence of a conducting singlet state formed by a localized dot degree of freedom and conduction electrons. Carbon nanotubes offer the possibility to study the emergence of the Kondo entanglement by tuning many-body correlations with a gate voltage. Here we show another side of Kondo correlations, which counterintuitively tend to block conduction channels: inelastic co-tunnelling lines in the magnetospectrum of a carbon nanotube strikingly disappear when tuning the gate voltage. Considering the global $SU(2) \otimes SU(2)$ symmetry of a nanotube coupled to leads, we find that only resonances involving flips of the Kramers pseudospins, associated to this symmetry, are observed at temperatures and voltages below the corresponding Kondo scale. Our results demonstrate the robust formation of entangled many-body states with no net pseudospin.

[1] Institute for Theoretical Physics, University of Regensburg, 93040 Regensburg, Germany. [2] Institut Néel, CNRS and Université Grenoble Alpes, 38042 Grenoble, France. [3] INAC-SPSMS, CEA and Université Grenoble Alpes, 38054 Grenoble, France. Correspondence and requests for materials should be addressed to M.G. (email: milena.grifoni@ur.de).

The ubiquity of Kondo resonances in quantum dots relies on the fact that their occurrence requires only the presence of degenerate dot states, whose degeneracy is associated with degrees of freedom that are conserved during the tunnelling on and out of the dot[1]. Finite magnetic fields can be used to break time-reversal symmetry-related degeneracies and unravel the deep nature of the Kondo state by tracking the magnetic field evolution of split Kondo peaks[2–11]. In a recent work[12], the striking report was made that specific transport resonances were not observable in nonlinear magnetoconductance measurements of split Kondo peaks in carbon nanotubes (CNTs), despite being expected from theoretical predictions[13–15]. Even more intriguing is that those resonances were recorded in inelastic co-tunnelling measurements in the weak-coupling regime[16]. Because in ref. 12 no comparative measurement for the weak-coupling regime was reported, the missing of resonances could not be unambiguously interpreted as a signature of the Kondo effect. From a closer inspection of other experimental reports for the Kondo regime[5,7,10,17], we notice that the absence of some resonances seems systematic.

In the following, we study the low-temperature nonlinear electron transport in a very clean CNT quantum dot[18]. By simply sweeping a gate voltage[8,19], we could tune the same CNT device from a weak-coupling regime, where Coulomb diamonds and inelastic co-tunnelling are observed, to a Kondo regime with strong many-body correlations to the leads. Then, using nonlinear magnetospectroscopy, transport resonances have been measured. The two regimes have been described using accurate transport calculations based on perturbative and nonperturbative approaches in the coupling, respectively. The missing resonances in the Kondo regime have been clearly identified, and their suppression fully taken into account by the transport theory. Accounting for both spin and orbital degrees of freedom, we discuss a global $SU(2) \otimes SU(2)$ symmetry related to the presence of two Kramers pairs in realistic CNT devices with spin–orbit coupling (SOC)[20–23] and valley mixing[16,21,24–26]. In virtue of an effective exchange interaction, virtual transitions that flip the Kramers pseudospins yield low-energy many-body singlet states with net zero Kramers pseudospin. This result in turn reveals that the transport resonances suppressed in the deep Kondo regime are associated with virtual processes that do not flip the Kramers pseudospin.

## Results

**Measurement and modelling of transport regimes.** The device under study consists of a semiconducting CNT, grown *in situ* on top of two platinum contacts, used as normal metal source and drain leads. Details of the device fabrication were reported previously[17] (see also the Methods). The CNT junction is suspended over an electrostatic gate and can be modelled as a single semiconducting quantum dot of size imposed by the contact separation ($\approx 200$ nm). All the measurements were performed at a mixing chamber temperature of about $T_{exp} = 30$ mK, which sets a lower bound to the actual electronic temperature. The set-up includes the possibility to fully rotate an in-plane magnetic field up to 1.5 T.

The CNT-level spectrum is depicted in Fig. 1a,b. Transverse bands, represented by the coloured hyperbolae in Fig. 1a, emerge from the graphene Dirac cones as a consequence of the quantization of the transverse momentum $k_\perp$. Bound states (bullets) are because of the quantization of the longitudinal momentum $k_\parallel$. Fourfold spin-valley degeneracy yields the exotic spin plus orbital $SU(4)$ Kondo effect[5,6,8,13,17,27,28]. The SOC removes the spin degeneracy of the transverse bands in the same valley (red and blue hyperbolae), and hence the $SU(4)$

symmetry[6,10,12,14,15,17,28,29]. Owing to the time-reversal symmetry, for each $k_\parallel$ a quartet of states consisting of two Kramers pairs splitted by the energy $\Delta = \Delta_{SO}$ arises. When also valley mixing is present, with the energy scale $\Delta_{KK'}$, orbital states are formed that are superpositions of valley states. A quartet now consists of two Kramers doublets at energies $\varepsilon_d = \pm \Delta/2$, with

$$\Delta = \sqrt{\Delta_{SO}^2 + \Delta_{KK'}^2},$$

see Fig. 1b.

By sweeping the gate voltage, the chemical potential is moved from above (electron sector) to below (hole sector) the charge neutrality point, and quadruplets of states are thus successively emptied. This pattern is visible in a typical measurement of the differential conductance $dI/dV$ versus the bias voltage $V_{sd}$ and the gate voltage $V_g$, Fig. 1c,d, which exhibits a characteristic fourfold periodicity. Figure 1c displays such a stability diagram for the electron sector, where Coulomb diamonds and inelastic co-tunnelling excitation lines are visible. Owing to significantly different ratios $\Gamma/U$ of the tunnel coupling to the charging energy in the valence and conduction regimes, Kondo physics dominates for odd hole number in the hole sector shown in Fig. 1d.

In order to investigate the dominant transport mechanisms, we have performed transport calculations for both regimes, using a standard minimal model for a longitudinal mode of a CNT quantum dot with SOC and valley mixing terms[16,18]. The explicit form of the model Hamiltonian $\hat{H}_{CNT}$ and the parameters used for the transport calculations are provided in the Methods. The transport calculations in the electron regime implement a perturbation theory that retains all tunnelling contributions to the dynamics of the CNT-reduced density matrix up to second order in the tunnel coupling $\Gamma$. This approximation thus accounts for Coulomb blockade (first order in $\Gamma$) and leading-order co-tunnelling processes (second order in $\Gamma$), and it is expected to give accurate results for small ratios $\Gamma/k_B T$ and $\Gamma/U$ (ref. 30). The results of the calculations for the differential conductance are shown in Fig. 1g—a gate trace in Fig. 1e. The perturbative theory reproduces the position of the inelastic co-tunnelling thresholds (panels 1c and 1g). In the gate trace of Fig. 1e, the experimental peaks are wider than the theoretical ones. Because in the latter the broadening is solely given by the temperature, this indicates that higher-order terms are responsible for a broadening of the order $\Gamma$ and for a Lamb shift of the experimental peaks[31–33]. In this work we are interested only in the evolution of the co-tunnelling resonances in magnetic field, which is well captured by the perturbative approach as long as Kondo ridges are not yet formed.

This situation radically changes in the hole sector where the gate trace reveals Kondo ridges for odd hole numbers. The theoretical trace in Fig. 1f is the outcome of a nonperturbative numerical density-matrix numerical renormalization group (DM-NRG) calculation[34] that uses the same model Hamiltonian but with slightly different parameters. The strong suppression of the conductance in the valley with even hole occupancy is an indication of the breaking of the $SU(4)$ symmetry in the presence of SOC and valley mixing to an $SU(2) \otimes SU(2)$ one[29,35]. In the DM-NRG calculations, the two-particles exchange $J$ was not included because of high computational costs. The latter further reduces the symmetry in the 2h valley (see, for example, the spectrum in Fig. 2b), and hence the experimental conductance is more rapidly suppressed in that valley than as predicted by our simulations. On the other hand, $J$ is not relevant for describing the spectrum in the 3h and 1h cases (Fig. 2a,c), which is the focus of the present work.

In the DM-NRG calculations, the fit to the experiment was performed assuming a temperature of $T = 30$ mK. From the so extracted parameters we evaluate the temperature dependence of the conductance at $-\varepsilon_d = U/2 - \Delta/2$, and $-\varepsilon_d = 5U/2 + \Delta/2$,

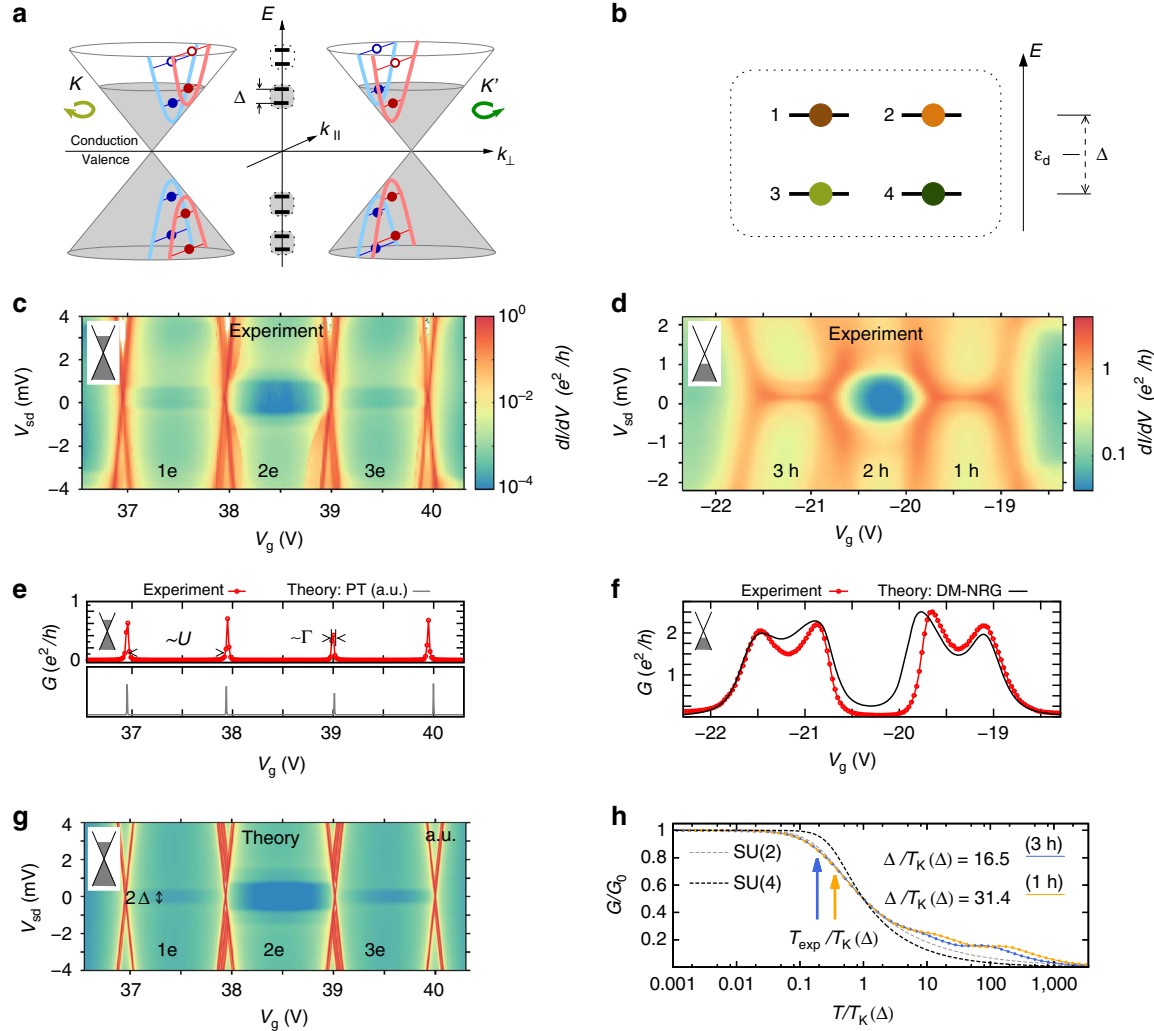

**Figure 1 | Transport regimes and bound states of a CNT quantum dot. (a)** A CNT with spin–orbit coupling is characterized by spin (blue, red) and valley (K, K′) resolved transverse modes (blue and red hyberbolae). The CNT chemical potential (upper limit of the shaded regions of the Dirac cones) is adjusted by sweeping the gate voltage from positive values (electron regime) to negative values (hole regime). Quantum confinement yields the quantization of the longitudinal momentum $k_{\parallel}$ (empty/solid bullets denote empty/filled bound states). **(b)** A generic quadruplet of bound states is composed of two Kramers doublets separated by the inter-Kramers splitting Δ. **(c,d)** Experimental stability diagrams demonstrating the successive filling of a quadruplet with electrons **(c)**, and holes **(d)**. On the electron side, sequential transport is exponentially suppressed inside the Coulomb valleys; the dominant mechanism is co-tunnelling. The appearance of high conductance ridges at zero bias **(d)** in valleys with odd holes is a signature of the Kondo effect. **(e,f)** Experimental gate traces at zero bias are compared with theoretical predictions obtained with perturbative **(e)** and nonperturbative DM-NRG **(f)** approaches. **(g)** Theoretical stability diagram for the electron side reproducing the experiment of **c**. **(h)** Scaling behaviour of the linear conductance in the middle of the valleys with odd hole numbers, $G_0 \approx 2e^2/h$. The system lies in the crossover regime ($0.1 < T_{exp}/T_K(\Delta) < 1$), as pointed out by the arrows. $T_K$ is the Kondo temperature determined from the DM-NRG calculation according to $G(T_K) = G_0/2$.

corresponding to gate voltage values located roughly in the middle of the 1h and 3h valleys, respectively, and extract the Kondo temperatures (see Fig. 1h). At such values of $\varepsilon_d$ the Kondo temperature takes its minimal value in a given valley, which sets a lower bound for $T_K$ (ref. 35). We find $T_K = 84$ mK and $T_K = 160$ mK for the 1h and 3h valleys, respectively. Correspondingly, $0.1 < T_{exp}/T_K < 1$, suggesting that the experiment is in the so-called Kondo crossover regime[1] also for the actual electronic and Kondo temperatures.

**Virtual transitions revealed by magnetospectroscopy.** Having set the relevant energy scales for both the electron and hole sectors, we proceed now with the investigation of magnetotransport measurements at finite source-drain bias, which have been performed for different fillings. A magnetic field **B** breaks

time-reversal symmetry and thus the Kramers degeneracies. By performing inelastic co-tunnelling spectroscopy, we can get information on the lowest lying resonances of our interacting system. The magnetospectrum corresponding to electron filling $n_e = 1, 2, 3$ of a longitudinal quadruplet, as expected for the perturbative regime, is shown in Fig. 2a–c. For the case of odd occupancies, we call $\mathcal{T}$ transition processes within a Kramers pair; $\mathcal{C}$ and $\mathcal{P}$ operations are associated to inter-Kramers transitions, as shown in Fig. 2a,c. Panels 2d–2f and 2g–2i show magnetotransport measurements and theoretical predictions for the electron and hole regimes, respectively. In these panels the current second derivative $d^2I/dV^2$ is reported. We have preferred this quantity over the more conventional $dI/dV$ (shown in the Supplementary Figs 4 and 5 and discussed in the Supplementary Note 4) to enhance eye visibility of the excitation spectra. In panels 2d–2f as well as 2h we have used our perturbative

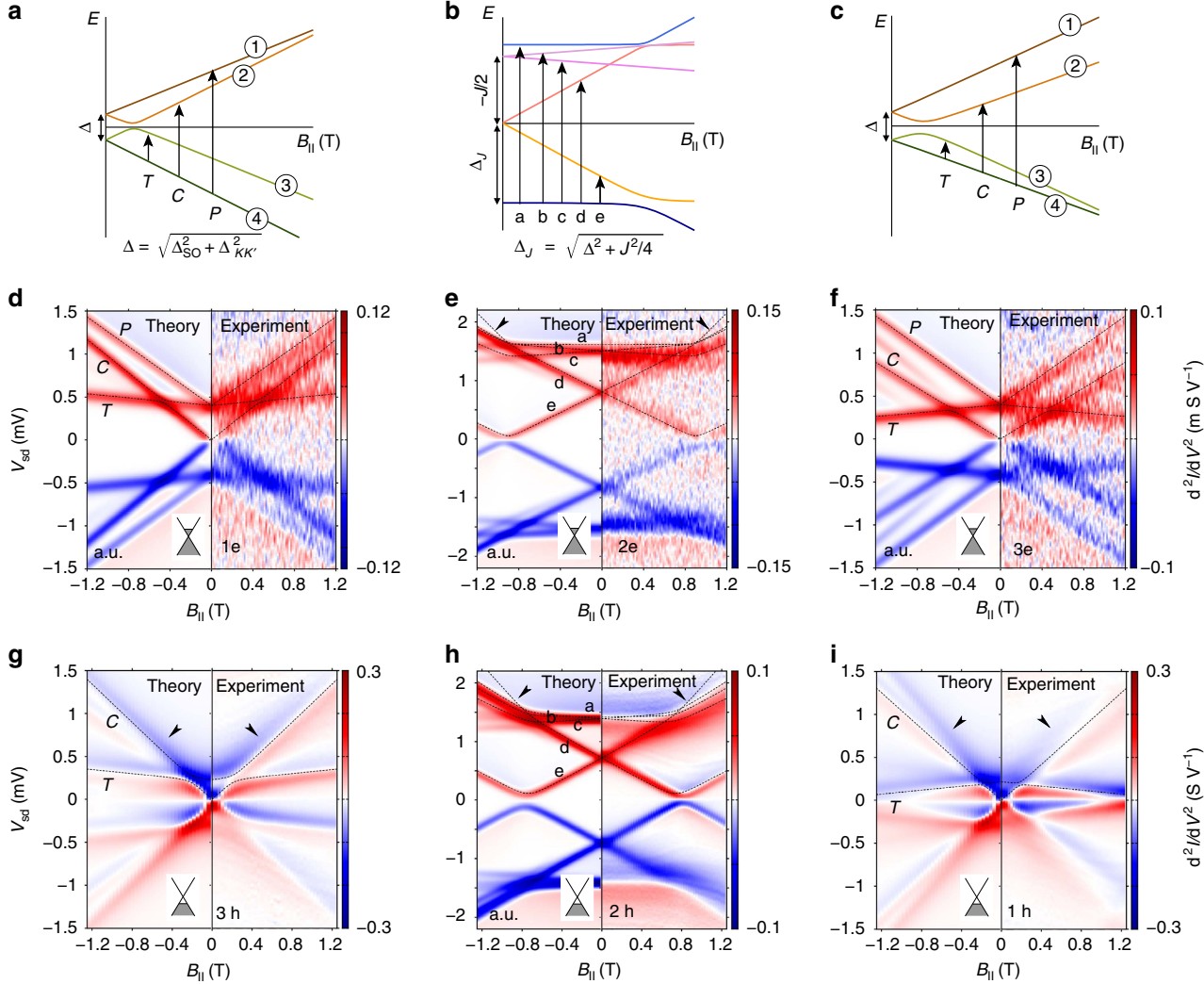

**Figure 2 | Energy spectra and magnetotransport in both co-tunnelling and Kondo regimes.** (**a–c**) Excitation spectra for electron filling ($n_e = 1, 2, 3$ from left to right). The parameters, $\Delta_{SO}$, $\Delta_{KK'}$ and $J$ account for SOC, valley mixing and exchange splitting, respectively. (**d–f**) Current second derivative $d^2I/dV^2$ in the electron regime at gate voltages fixed in the middle of the 1e, 2e and 3e charge states, as a function of bias voltage and parallel magnetic field. Each panel reports experimental data (positive magnetic field) and transport calculations (negative field). The dotted lines correspond to the transition energies from the ground state calculated directly from the spectra (**a–c**). At odd filling (**a,c**), all possible ground-state transitions, denoted by $\mathcal{C}$, $\mathcal{P}$ and $\mathcal{T}$, are observed in the experiment (**d,f**). Being signalled by co-tunnelling steps in the current first derivative, they yield maxima/minima in the second derivative. Likewise for even occupation, except for the 'a' transition at high field (marked by arrows), forbidden by selection rules. (**g–i**) $d^2I/dV^2$ maps in the hole regime for the 1h, 2h and 3h charge states. While the experimental results for the 2h and 2e cases are similar, the $\mathcal{P}$ transitions are no longer experimentally resolved, as predicted by the transport theory because of the Kondo effect (**g,i**). These missing resonances are indicated by arrows in **g,i**. In the Kondo regime $\mathcal{T}$ and $\mathcal{C}$ transitions yield maxima in the differential conductance, and hence zeroes in the second derivative. Near maxima (minima) of $dI/dV$ the second derivative decreases (increases), that is, it changes from red to blue (blue to red) upon increasing the bias. The experimental part of **g–i** has been adapted from[17].

approach[30]. The calculations in Fig. 2g,i, in contrast, are based on the Keldysh effective action (KEA) method[36,37] and are nonperturbative. The nature of the dominant inelastic transitions is clearly identified by simply looking at the excitation spectrum (dashed lines in Fig. 2d–i). All inelastic transitions from the ground state are resolved in the co-tunnelling spectroscopy performed in the low coupling electron regime, similar to previous reports[16]. When inspecting the hole regime, it is clear that only for the 2h case, panel 2h, the experimental data can be interpreted by means of a simple co-tunnelling excitation spectrum; moreover, the 2e and 2h co-tunnelling spectra are very similar. In the 1h and 3h cases shown in panels 2g, 2i Kondo correlations dominate the low-energy transport, and differences with respect to the electron sector are seen. The

zero-bias Kondo peak does not immediately split as the field is applied; rather the splitting occurs at a critical field such that the energy associated to the inelastic $\mathcal{T}$ transition is of the order of the Kondo temperature[1]. In the 1 h valley, the lowest pair of levels merges again for values of the field of ~1.2 Tesla, yielding a Kondo revival[5,29]. Bias traces of the differential conductance highlighting the revival are shown in the Supplementary Fig. 3 and analysed in the Supplementry Note 3. Striking here is the observation that, in contrast to the 1e and 3e cases, only one of the two inter-Kramers transitions is resolved in the experimental data for the 3h and 1h valley. However, in particular for the 1h case, the $\mathcal{P}$ and $\mathcal{C}$ excitation lines, as expected from the excitation spectrum, should be separated enough to be experimentally distinguishable, similar to the 3e case. By comparing with the

excitation spectrum (dashed lines in panels 2g, 2i), we conclude that it is the $\mathcal{P}$ transition, which is not resolved. Our KEA transport theory qualitatively reproduces these experimental features.

Magnetotransport measurements performed for other quadruplets both in the conduction and valence regimes exhibit qualitatively similar features (see Supplementary Figs 6–8, Supplementary Table 1 and Supplementry Note 5), and hence confirm the robustness of the suppression of $\mathcal{P}$ transitions in the Kondo regime. Our results naturally reconcile the apparently contradictory observations in refs 12,16. Furthermore, they suggest that the inhibition of selected resonances in the Kondo regime is of fundamental nature.

**Fundamental symmetries of correlated CNTs.** To understand the experimental observations microscopically, we have analysed those symmetries of an isolated CNT, which also hold in the presence of on-site Coulomb repulsion typical of Anderson models.

In the absence of a magnetic field, one finds a $U(1) \otimes U(1) \otimes SU(2) \otimes SU(2)$ symmetry related to the existence of two pairs of time-reversal degenerate doublets (see Fig. 1b) called in the following upper ($u$) and lower ($d$) Kramers channels. The $U(1)$ symmetries reflect charge conservation in each Kramers pair with generators $\hat{Q}_\kappa = \frac{1}{2}\sum_{j\in\kappa}\left(\hat{n}_j - \frac{1}{2}\right)$, which measure the charge of the pair with respect to the half-filling. Here is $j = (1, 2)$ or $(3, 4)$ for $\kappa = u$ or $d$. The $SU(2)$ symmetries are generated by the spin-like operators $\hat{\mathbf{J}}_\kappa = \frac{1}{2}\sum_{j,j'\in\kappa}\hat{d}_j^\dagger \boldsymbol{\sigma}_{j,j'}\hat{d}_{j'}$. Here $\boldsymbol{\sigma}$ is the vector of Pauli matrices. Physically, $\hat{J}_u^z = (\hat{n}_1 - \hat{n}_2)/2$ and $\hat{J}_d^z = (\hat{n}_4 - \hat{n}_3)/2$ account for the charge unbalance within the Kramers pair. Thus, an isolated CNT with one electron or a hole only in the quadruplet has a net Kramers pseudospin (and charge). Figure 3a shows the two degenerate ground-state configurations $|\Downarrow; -\rangle$, $|\Uparrow -\rangle$ of the isolated CNT with an unpaired effective spin ($\Downarrow$ or $\Uparrow$) in the lowest Kramers pair and no occupation (symbol '$-$') of the upper Kramers pair. In the weak-coupling regime, a perturbative approach to linear transport accounts for elastic co-tunnelling processes involving the doubly degenerate ground-state pair[38].

These virtual transitions are denoted $\mathcal{I}$ or $\mathcal{T}$ when they involve the same state or its Kramers partner, respectively (see Fig. 3a). A finite magnetic field breaks the $SU(2)$ symmetries. However, former degenerate CNT states can still be characterized according to the eigenvalues of the $\hat{Q}_\kappa$ and $\hat{J}_\kappa^z$ operators, since they commute with the single-particle CNT Hamiltonian, which has in the Kramers basis the form (see Methods):

$$\hat{H}_0 = \sum_{\kappa=\pm}\left(\bar{\varepsilon}(\mathbf{B}) + \kappa\frac{\bar{\Delta}(\mathbf{B})}{2}\right)\hat{N}_\kappa + 2\delta\varepsilon(\mathbf{B}) + \kappa\delta\Delta(\mathbf{B})\hat{J}_\kappa^z, \quad (1)$$

where $u/d = +/-$, $\hat{N}_\kappa = 2\hat{Q}_\kappa + 1$, and at zero field is $\bar{\Delta}(B=0) = \Delta$, $\bar{\varepsilon}(B=0) = \varepsilon_d$, $\delta\varepsilon = \delta\Delta = 0$. Hence, our finite bias and finite magnetic field spectroscopy allows us to clearly identify the relevant elastic and inelastic virtual processes according to the involved Kramers charge and spin. As illustrated in Fig. 3b, in the weak tunnelling regime only energy differences matter in our model, and hence both intra-Kramers ($\mathcal{I}$, $\mathcal{T}$) and inter-Kramers ($\mathcal{P}$, $\mathcal{C}$) transitions are expected in transport. In the Kondo regime this picture changes. As we shall demonstrate, emerging Kondo correlations lead to the progressive screening of the Kramers pseudospin of the dot by the conduction electrons.

To this aim we observe that, when a sizeable tunnel coupling to the leads is included, the CNT charge and pseudospin operators $\hat{Q}_\kappa$ and $\hat{J}_k$ are no longer symmetries of the coupled system, since the tunnelling does not conserve the dot particle number. The occurrence of the Kondo effect, however, suggests that the CNT quantum numbers $j = 1, 2, 3, 4$ are carried also by the conduction electrons and conserved during tunnelling[13]. This is the case when the dot is only a segment of the CNT (see Supplementary Fig. 1). Following ref. 35, we hence introduce charge, $\hat{\mathcal{Q}}_\kappa = \hat{Q}_\kappa + \hat{Q}_{L,\kappa}$, and pseudospin, $\hat{\boldsymbol{\mathcal{J}}}_\kappa = \hat{\mathbf{J}}_\kappa + \hat{\mathbf{J}}_{L,\kappa}$, operators of the coupled CNT plus leads (L) system. Under the assumption that the tunnelling couplings are the same within each Kramers channel $\kappa = u,d$, the total Hamiltonian (see Supplementary Methods) commutes with the charge and pseudospin operators $\hat{\mathcal{Q}}_\kappa$ and $\hat{\boldsymbol{\mathcal{J}}}_\kappa$, which hence generate a $U(1) \otimes U(1) \otimes SU(2) \otimes SU(2)$ symmetry of the coupled system. As a consequence, many-body states can be characterized by the quadruplet of eigenvalues

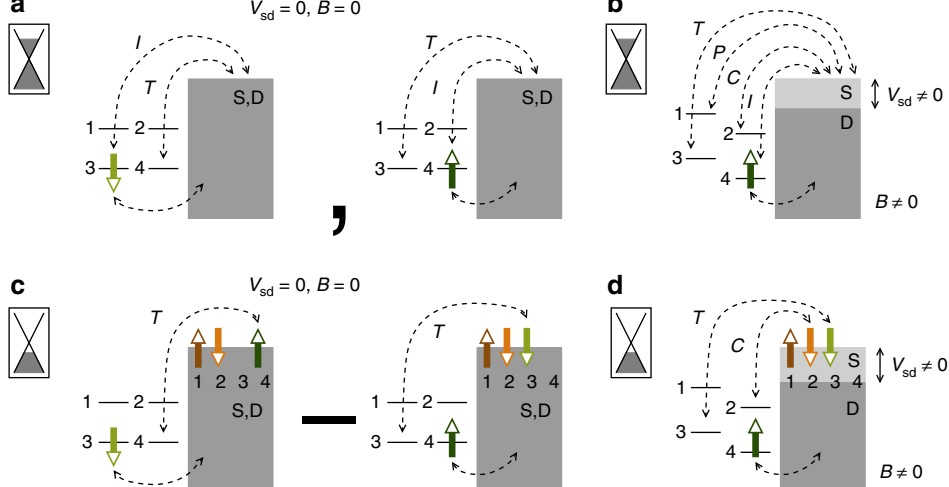

**Figure 3 | Ground-state configurations and virtual processes of a CNT quantum dot with one-electron filling in the co-tunnelling and Kondo regimes.** (**a**) In the co-tunnelling regime the one-electron ground state is doubly degenerate, with opposite values of the Kramers pseudospin. Elastic co-tunnelling processes to source (S) and drain (D) leads (grey areas) involving the same pseudospin, $\mathcal{I}$, and its Kramers partner, $\mathcal{T}$, contribute to the linear transport. (**b**) Kramers degeneracy is broken by a magnetic field. A finite bias allows us to identify the three inelastic processes $\mathcal{T}$, $\mathcal{P}$ and $\mathcal{C}$, which connect the bound states within a quadruplet. (**c**) The ground state in the Kondo regime is a singlet with no net Kramers pseudospin. Virtual $\mathcal{T}$ fluctuations which involve a pseudospin flip dominate at low energies. (**d**) At finite bias voltages the inelastic $\mathcal{T}$, $\mathcal{C}$ transitions, which involve a pseudospin flip, are the most relevant in the deep Kondo regime.

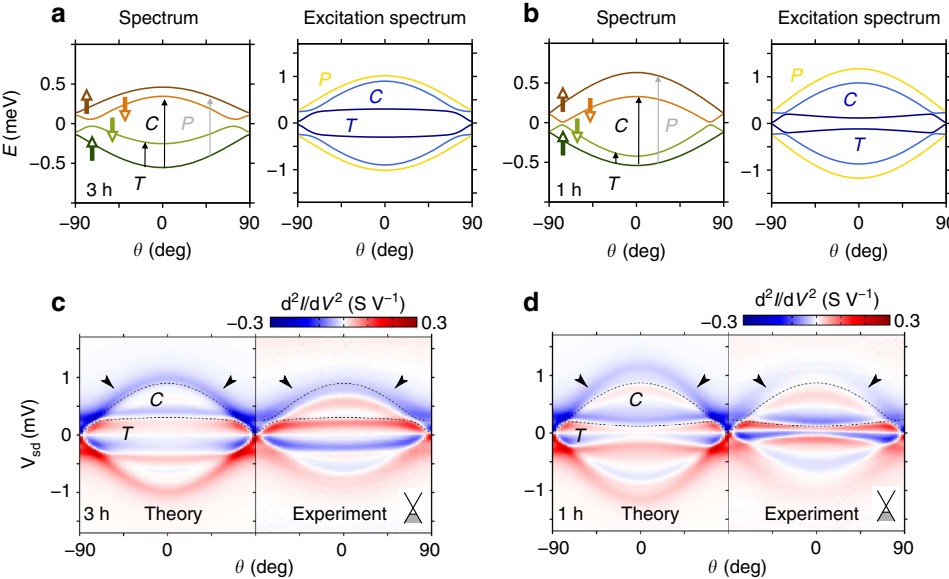

**Figure 4 | Angular dependence of both spectrum and transport characteristics as the magnetic field is rotated in the CNT plane.** (**a,b**) Sketch of the spectrum and excitation spectrum at 3h and 1h fillings, respectively, as a function of the polar angle $\theta$ formed by an applied magnetic field and the CNT axis. A classification of the inelastic transitons according to the $\mathcal{T}$, $\mathcal{C}$ and $\mathcal{P}$ operations is still possible. (**c,d**) As-measured and KEA transport calculations for the current's second derivative $d^2I/dV^2$. The absence of $\mathcal{P}$ transitions is independent of the direction of the applied field. The experimental part of **d** has been adapted from ref. 17. The magnetic field magnitude in **a–d** is set to 0.8 T.

$(\mathcal{Q}_d, \mathcal{Q}_u; \mathcal{J}_d, \mathcal{J}_u)$, where the highest eigenvalue $\mathcal{J}_\kappa$ of $\hat{\mathcal{J}}_\kappa^z$ is indicated in the quadruplet. This notation gives direct access to the eigenvalues $\mathcal{J}_\kappa(\mathcal{J}_\kappa+1)$ of $\hat{\mathcal{J}}_\kappa^2$. Such quadruplets can be numerically calculated within our scheme for the Budapest DM-NRG code[39], and yield (for the valleys with one electron or one hole) a *singlet* ground state characterized by the quadruplet $(0, 0; 0, 0)$. Thus, '0' is also eigenvalue of $\hat{\mathcal{J}}_d^2$ and $\hat{\mathcal{J}}_u^2$. That is, we find a unique ground state with no net pseudospin. This situation is illustrated in Fig. 3c: because of $\mathcal{Q}_k=0$, the Kramers channels are half-filled (two charges per channel), whereby one charge arises from the electron trapped in the CNT itself. For $\Delta=0$ this CNT charge is equally distributed among the two channels, while for large values of $\Delta/T_K(\Delta)$, as in our calculation (see Fig. 1h), it is mainly in the lowest Kramers channel. Thus, at zero temperature the localized CNT pseudospin is fully screened by an opposite net pseudospin in the leads. In the orthonormal basis $\{|m\rangle \otimes |n\rangle_L\}$, spanned by the pseudospin eigenstates of CNT and leads, this ground state is characterized by the entangled configuration $\frac{1}{\sqrt{2}}[|\Uparrow; -\rangle \otimes |\Downarrow; \Downarrow, \Uparrow\rangle_L - |\Downarrow; -\rangle \otimes |\Uparrow; \Downarrow, \Uparrow\rangle_L]$ of dot and leads pseudospins.

In the standard spin-1/2 Kondo effect, the appearance of a unique singlet ground state with no net spin is the result of the screening of the quantum impurity spin by the conduction electron spins because of the antiferromagnetic character of the coupling constant between such degrees of freedom[1]. Triplets are excited states of the system. To interpret the spin-1/2 Kondo effect in quantum dots, it is possible to derive from an Anderson model an effective Kondo Hamiltonian[40] given by the product of the quantum dot spin and the conduction electron spin. The coupling constant for this product is positive and thus antiferromagnetic. In addition, for the more complex case of a CNT, effective Kondo Hamiltonians have been derived, with positive coupling constants for Kramers channels identified by orbital and spin degrees of freedom[13,41]. The antiferromagnetic character of the coupling constants remains also when, as in our case, the more abstract Kramers pseudospin is used.

A natural consequence of the antiferromagnetic nature of the correlations is that at low temperatures and zero-bias elastic virtual transitions, which flip the pseudospin, that is, $\mathcal{T}$ transitions, are favoured, as depicted in Fig. 3c. Similarly, $\mathcal{C}$ transitions are inelastic processes that flip the pseudospin and become accessible at finite bias, as shown in Fig. 3d. They connect the singlet ground state to an excited state where the CNT charge is located in the upper Kramers channel. Our results suggest that $\mathcal{P}$ transitions are inhibited because they involve virtual transitions that conserve the pseudospin.

**Entanglement of Kramers pseudospins.** To further confirm that it is the Kramers pseudospins and not distinct spin or orbital degrees of freedom, which should be considered in the most general situations, we report results for the differential conductance as a function of the angle $\theta$ formed by the magnetic field and the CNT's axis. The combined action of SOC, valley mixing and non-collinear magnetic field mixes spin and valley degrees of freedom which, in general, are no longer good quantum numbers to classify CNT states. Nevertheless, the three discrete $\mathcal{T}$, $\mathcal{P}$ and $\mathcal{C}$ operations still enable us to identify the inelastic transitions in the 1h and 3h case, independent of the direction of the magnetic field. The angular dependence of both energy and excitation spectra for a fixed magnetic field amplitude is shown in Fig. 4a,b for the 3h and 1h fillings, respectively. The corresponding transport spectra are shown in Fig. 4c,d, respectively. A perpendicular magnetic field almost restores (for our parameter set) Kramers degeneracy, thus revitalizing the Kondo resonance for this angle. As the field is more and more aligned to the CNT's axis, the degeneracy is removed, which also enables us to distinguish between $\mathcal{P}$ and $\mathcal{C}$ transitions. As in the axial case of Fig. 2, only the inelastic resonance associated with the $\mathcal{C}$ transition is clearly resolved in both the experiment and theory.

**Entropy and specific heat**. Usually, quantum entanglement suffers from decoherence effects[42,43]. The Kondo–Kramers singlets, however, are associated with a global symmetry of the quantum dot-plus lead complex, and are robust against thermal fluctuations or finite bias effects as long as the impurity is in the

## Table 1 | Parameter set.

| | Holes (shell $N_h = 6$) | Electrons (shell $N_e = 6$) |
|---|---|---|
| $\Delta_{SO}$ (meV) | $-0.21$ | $-0.4$ |
| $\Delta_{KK'}$ (meV) | $0.08$ | $0.04$ |
| $\mu_{orb}$ (meV per T) | 0.51 (3h), 0.51 (2h), 0.55 (1h) | 0.43 |
| $U$ (meV) PT | | 26,5 |
| $U$ (meV) NRG | 4,7 | |
| $U$ (meV) KEA | $\infty$ (3h, 1h) | |
| $J$ (meV) PT | $-1.35$ | $-1.4$ |
| $\Delta_\mu B_{\parallel}$ (meV per T) | $-0.05$ | $-0.06$ |
| $e\Delta V_{sd}$ (meV) | $0.12$ | $0.28$ |

CNT, carbon nanotube; KEA, Keldysh effective action; NRG, numerical renormalization group; PT, perturbation theory.
The table shows the parameters used to fit the electronic transport spectra of the CNT in the gate voltage region shown in the main text. It corresponds to the valence quadruplet $N_h = 6$ (hole transport), and the conduction quadruplet $N_e = 6$ (electron transport), counting the Coulomb diamonds from the band gap. PT, NRG and KEA refer to the three theoretical methods used in our calculations (see text). The experimental data for each Coulomb valley are offset by $\Delta V_{sd}$, and tilted in the magnetic field by $\Delta_\mu B_{\parallel}$, resulting in an asymmetry between the measurement in fields parallel and antiparallel to the CNT axis. In all the plots presented in the work both the offset and the tilt have been removed.

Fermi liquid regime[1] ($T < 0.01\ T_K$ for our experiment). For larger energy scales, $0.01 < T/T_K < 1$ the impurity is not fully screened, but Kondo correlations persist yielding universal behaviour of relevant observables, as seen, for example, in Fig. 1h at the level of the linear conductance. In order to further investigate the impact of thermal fluctuations on Kondo correlations, we have calculated the temperature dependence of the impurity entropy $S_{CNT} = S_{tot} - S_L$, where the $S_i$ is thermodynamic entropy, and of the impurity-specific heat[44] (see Supplementry Note 1 and Supplementary Fig. 2). The conditional entropy $S_{CNT}(T)$ remains close to zero up to temperatures $T \approx 0.01\ T_K$, indicating that the system is to a good approximation in the singlet ground state. At higher temperatures, the impurity entropy grows, but universality is preserved up to temperatures close to $T_K$, at which the entropy approaches the value $k_B \log 2$.

## Discussion

Our results show that specific low-energy inelastic processes, observed in the perturbative co-tunnelling regime, tend to be blocked in the Kondo regime because of antiferromagnetic-like correlations, which at zero temperature yield a many-body ground state with net zero Kramers pseudospin. This signature of the Kondo effect is universal in the sense that it does not depend on the degree of the SOC or valley mixing specific to a given CNT. As such, it is also expected for $SU(4)$-correlated CNTs, which explains the missing inelastic resonance in the seminal work[5]. Furthermore, we believe that such pseudospin-selective suppression should be detectable also in a variety of other tunable quantum dot systems with emergent $SU(4)$ and $SU(2) \otimes SU(2)$ Kondo effects[4,11,45–48].

Because the screening is progressively suppressed by increasing the temperature or the bias voltage, it should be possible to recover such inelastic transitions by continuously tuning those parameters. Indeed, signatures of the re-emergence of the $\mathcal{P}$ transition are seen in the KEA calculations and experimental traces at fields $\sim 0.9\ T$ in the form of an emerging shoulder (see Supplementary Fig. 3). Experiments at larger magnetic fields, not accessible to our experiment, are required to record the evolution of this shoulder, and thus the suppression of (non-equilibrium) Kondo correlations by an applied bias voltage.

## Methods

**Experimental fabrication.** Devices were fabricated from degenerately doped silicon Si/SiO$_2$/Si$_3$N$_4$ wafers with a 500 nm-thick thermally grown SiO$_2$ layer and 50 nm Si$_3$N$_4$ on top. Metal leads separated by 200 nm were first defined by electron-beam lithography and deposited using electron-gun evaporation. A thickness of 2 nm Cr followed by 50 nm Pt was used. A 200 nm-deep trench was created using both dry-etching and wet-etching. A second step of electron-beam lithography was used to design a 50 nm-thin metallic local gate at the bottom of the trench. Catalyst was then deposited locally on top of the metal leads. CNTs were then grown by the carbon vapor deposition (CVD) technique to produce as clean as possible devices. Only devices with room temperature resistances below 100 k$\Omega$ were selected for further studies at very low temperature. A scanning electron microscopy of a device similar to the one measured in this work is shown in the Supplementary Fig. 1.

**Transport methods.** For the transport calculations, three different approaches have been used: the DM-NRG method, a real-time diagrammatic perturbation theory for the dynamics of the reduced density and the analytical KEA approach. Further details are discussed in the Supplementary Note 2.

**Model CNT Hamiltonian.** In our calculations we have used the standard model Hamiltonian for the longitudinal mode of a CNT accounting for SOC, valley mixing, on-site and exchange Coulomb interactions and an external magnetic field[18]. Regarding both SOC and the valley mixing as perturbations breaking the $SU(4)$ symmetry of the single-particle CNT Hamiltonian, it has the general form

$$\hat{H}_{CNT} = \hat{H}_d + \hat{H}_{SO} + \hat{H}_{KK'} + \hat{H}_U + \hat{H}_J + \hat{H}_B, \quad (2)$$

where $\hat{H}_d + \hat{H}_U$ is the $SU(4)$ invariant component. In the basis set $\{K' \uparrow, K' \downarrow, K \uparrow, K \downarrow\}$ indexed by the valley and spin degrees of freedom $\tau = K', K = \pm$ and $\sigma = \uparrow, \downarrow = \pm$, respectively, it reads

$$\hat{H}_d + \hat{H}_U = \varepsilon_d \sum_{\tau,\sigma=\pm} \hat{d}^\dagger_{\tau,\sigma} \hat{d}_{\tau,\sigma} + \frac{U}{2} \sum_{(\tau,\sigma) \neq (\tau',\sigma')} \hat{n}_{\tau,\sigma} \hat{n}_{\tau',\sigma'}, \quad (3)$$

with $\varepsilon_d$ being the energy of the quantized longitudinal mode, which can be tuned through the applied gate voltage and $U$ accounting for charging effects. Valley mixing and SOC break the $SU(4)$ symmetry with characteristic energies $\Delta_{KK'}$ and $\Delta_{SO}$, respectively. The corresponding contributions read:

$$\hat{H}_{KK'} + \hat{H}_{SO} = \frac{\Delta_{KK'}}{2} \sum_{\tau,\sigma=\pm} \hat{d}^\dagger_{\tau,\sigma} \hat{d}_{-\tau,\sigma} + \frac{\Delta_{SO}}{2} \sum_{\tau,\sigma=\pm} \sigma\tau \hat{n}_{\tau,\sigma}. \quad (4)$$

The SOC term is a result of the atomic spin–orbit interaction in carbon, and thus exists also for ideally infinitely long CNTs[20]. The valley mixing, in contrast, is absent in long and defect-free CNTs. It only arises because of scattering off the boundaries in finite-length CNTs or because of disorder[21,25,26]. It is expected to be zero in disorder-free CNTs of the zig-zag class, according to angular momentum conservation rules, and finite in CNTs of the armchair class[26]. In our experiments, according to Table 1, the valley mixing is very small, which suggests a tube of the zig-zag class.

Similar to the SOC and valley mixing, the exchange interaction preserves time-reversal symmetry. Its microscopic form is not known for arbritary chiral angles. It has been evaluated so far for the case of pure armchair tubes[49], and for the zig-zag class[18,50] CNTs. Because the experiments suggest that our tube is of the zig-zag class, we choose in the following a form suitable to describe this case. It reads

$$\hat{H}_J = -\frac{J}{2} \sum_{\sigma=\pm} \left\{ \hat{n}_{K,\sigma} \hat{n}_{K',\sigma} + \hat{d}^\dagger_{K,\sigma} \hat{d}^\dagger_{K',-\sigma} \hat{d}_{K,-\sigma} \hat{d}_{K',\sigma} \right\}, \quad (5)$$

with $J < 0$ the exchange coupling. Finally, contributions arising from a magnetic field $\mathbf{B}$ contain both Zeeman and orbital parts. Decomposing $\mathbf{B}$ into components parallel and perpendicular to the tube axis, $B_\parallel = B\cos\theta$ and $B_\perp = B\sin\theta$, respectively, one finds:

$$
\begin{aligned}
\hat{H}_B &= \hat{H}_B^Z + \hat{H}_B^{orb} \\
&= B_\parallel \sum_{\tau,\sigma=\pm} \left(\frac{g}{2}\mu_B \sigma + \mu_{orb}\tau\right) \hat{d}^\dagger_{\tau,\sigma} \hat{d}_{\tau,\sigma} \\
&\quad + \frac{g}{2}\mu_B B_\perp \sum_{\tau,\sigma=\pm} \hat{d}^\dagger_{\tau,\sigma} \hat{d}_{\tau,-\sigma}.
\end{aligned} \quad (6)
$$

Notice that the spin and valley remain good quantum numbers in the presence of an axial field ($\theta = 0, \pi$), while a perpendicular component flips the spin degrees of freedom. The parameters of the CNT Hamiltonian used to fit the experimental data shown in Figs 1, 2 and 4 are listed in Table 1.

**Kramers charge and pseudospin representation.** We call Kramers basis the quadruplet $\{|i\rangle\}$, $i = 1, 2, 3, 4$ (shown in Fig. 1b), which diagonalizes the single-particle part $\hat{H}_0 = \hat{H}_d + \hat{H}_{KK'} + \hat{H}_{SO} + \hat{H}_B$ of the CNT Hamiltonian. For magnetic fields parallel or perpendicular to the CNT axis, this Hamiltonian is easily diagonalized, see, for example, ref. 12. For other orientations of the field, because of the combined action of SOC and valley mixing, such states are a linear superposition of all the basis states $\{|\tau,\sigma\rangle\}$, such that neither the spin nor the valley are in general good quantum numbers any more. One has to resort to

numerical tools to find both the eigenvectors $\{|i\rangle\}$ and the eigenvalues $\varepsilon_i$, $i = 1, 2, 3, 4$. The angular dependence of these eigenenergies is sketched in Fig. 4.

Despite the complexity inherent in the Hamiltonian $\hat{H}_0$, a closer inspection reveals the existence of conjugation relations among the quadruplet of states $i = 1, 2, 3, 4$ generated by the time-reversal operator $\hat{T}$, as well as by the particle–hole-like and chirality operators $\hat{P}$ and $\hat{C} = \hat{P}\hat{T}^{-1}$, respectively[12]. Specifically, the states are ordered such that $(1, 2)$ and $(3, 4)$ are time-reversal partners, while $(1, 4)$ and $(2, 3)$ are particle–hole partners. In the $\{|\tau, \sigma\rangle\}$ basis the operators read

$$\hat{T} = -\hat{\kappa} \sum_{\tau,\sigma} \sigma \hat{d}^{\dagger}_{-\tau,-\sigma} \hat{d}_{\tau,\sigma}, \tag{7}$$

$$\hat{P} = \hat{\kappa} \sum_{\tau,\sigma} \sigma\tau \hat{d}^{\dagger}_{-\tau,\sigma} \hat{d}_{\tau,\sigma}, \tag{8}$$

$$\hat{C} = \sum_{\tau,\sigma} \tau \hat{d}^{\dagger}_{\tau,-\sigma} \hat{d}_{\tau,\sigma}, \tag{9}$$

where $\hat{\kappa}$ stands for the complex conjugation operator. In the absence of a magnetic field $\hat{T}$ commutes with the total CNT Hamiltonian, yielding a single-particle spectrum with two degenerate Kramers doublets $(1, 2)$ and $(3, 4)$ separated by the inter-Kramers splitting $\Delta = \sqrt{\Delta_{SO}^2 + \Delta_{KK'}^2}$ (see Fig. 1b). As far as the $\hat{P}$ and $\hat{C}$ operators are concerned, at zero magnetic field they are symmetries only in the absence of SOC and valley mixing. Since both anticommute with $\hat{H}_{SO} + \hat{H}_{KK'}$, it holds for $\mathcal{P}$-conjugated pairs, $\varepsilon_{1,2}(\Delta) = \varepsilon_{4,3}(-\Delta)$. A magnetic field breaks the time-reversal symmetry; however, because $\hat{H}_B$ anticommutes with $\hat{T}$, formerly degenerate Kramers states are still related to each other by Kramers conjugation. For an arbitrary magnetic field **B** time-reversal conjugation and particle–hole conjugation imply[12]:

$$\varepsilon_{1,4}(\mathbf{B}) = \varepsilon(\mathbf{B}) \pm \frac{1}{2}\Delta(\mathbf{B}), \tag{10}$$

$$\varepsilon_{2,3}(-\mathbf{B}) = \varepsilon_{1,4}(\mathbf{B}), \tag{11}$$

where $\varepsilon(\mathbf{B})$ and $\Delta(\mathbf{B})$ reduce to the longitudinal energy and Kramers splitting $\varepsilon_d$ and $\Delta$, respectively, at zero field.

These relations clearly suggest the introduction of auxiliary charge $\hat{N}_{ij} := \hat{n}_i + \hat{n}_j$ and pseudospin $\hat{J}^z_{ij} = (\hat{n}_i - \hat{n}_j)/2$ operators, in terms of which we can write

$$\hat{H}_0 = \varepsilon(\mathbf{B})\hat{N}_{14} + \Delta(\mathbf{B})\hat{J}^z_{14} + \varepsilon(-\mathbf{B})\hat{N}_{23} + \Delta(-\mathbf{B})\hat{J}^z_{23}.$$

Introducing the average quantities $\bar{\Delta}(\mathbf{B}) := (\Delta(\mathbf{B}) + \Delta(-\mathbf{B}))/2$, $\bar{\varepsilon}(\mathbf{B}) := (\varepsilon(\mathbf{B}) + \varepsilon(-\mathbf{B}))/2$, as well as the differences $\delta\Delta(\mathbf{B}) := (\Delta(\mathbf{B}) - \Delta(-\mathbf{B}))/2$, $\delta\varepsilon(\mathbf{B}) := (\varepsilon(\mathbf{B}) - \varepsilon(-\mathbf{B}))/2$, the CNT Hamiltonian can be easily recast in terms of total charge and pseudospin of a Kramers pair. It reads:

$$\hat{H}_0 = \left(\bar{\varepsilon}(\mathbf{B}) + \frac{\bar{\Delta}(\mathbf{B})}{2}\right)\hat{N}_{12} + [2\delta\varepsilon(\mathbf{B}) + \delta\Delta(\mathbf{B})]\hat{J}^z_{12}$$
$$+ \left(\bar{\varepsilon}(\mathbf{B}) - \frac{\bar{\Delta}(\mathbf{B})}{2}\right)\hat{N}_{43} + [2\delta\varepsilon(\mathbf{B}) - \delta\Delta(\mathbf{B})]\hat{J}^z_{43}.$$

Such equation is equation (1) in the main part of the manuscript upon calling $\hat{J}^z_{43} = \hat{J}^z_d$, $\hat{J}^z_{12} = \hat{J}^z_u$, and similarly $\hat{N}_{43} = \hat{N}_d$, $\hat{N}_{12} = \hat{N}_u$.

**Data availability.** The data that support the main findings of this study are available from the corresponding author upon request.

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

## Acknowledgements

We acknowledge fruitful discussions with C. Strunk, A. Hüttel, G. Zaránd and C. P. Moca as well as the financial support by the Deutsche Forschungsgemeinschaft via SFB 689 and GRK 1570, and by the ERC Advanced Grant MolNanoSpin No. 226558.

## Author contributions

M.N. performed the perturbative non-equilibrium calculations, S.S. evaluated the differential conductance in the Kondo regime using the non-equilibrium KEA approach, while D.M. did the equilibrium DM-NRG simulations. M.M. evaluated the magnetospectrum of the isolated nanotube and devised all the figures. N.-V.N. helped to fabricate and characterize the devices, J.-P.C. carried out and analysed the experiments, while W.W. supervised them. M.G. performed the theoretical analysis and wrote the manuscript with critical comments provided by all authors.

## Additional information

**Competing financial interests:** The authors declare no competing financial interests.

