## [Peer review file · Nature Communications]

Reviewers' Comments:

Reviewer #1 (Remarks to the Author)

In this paper, the authors consider a clean carbon nanotube quantum dots and present a detailed analysis of magneto-transport properties. The authors convincingly show that some inelastic cotunneling spectral lines in the magnetospectrum disappear when the nanotube quantum dot is tuned in the Kondo regime using a gate voltage. The data are very clean with the highest possible resolution which allows the authors to potentially visualize all inelastic cotunneling lines. These experimental data are qualitatively and almost quantitatively explained by a combination of sophisticated theoretical approaches applied on a simple paradigmatic model involving a SU(4) Anderson Hamiltonian.

The whole paper is very nicely and clearly written and the figures clearly illustrates the physics at play and the agreement theory/experiment is quite remarkable.

However, there are a few points I would like the authors to clarify both on the analysis and on the presentation.

- In Fig 1f, the agreement between the DM NRG calculations and the data is only qualitative. What are the origins of such discrepancies. Such equilibrium analysis is important since this should to fix/determine the microscopic parameters of the Anderson model. Along the same line, why the authors should use different parameters for the NRG approach compared to other methods.

- In order to describe the data, the authors assume that the CNT quantum numbers are good quantum numbers i.e. are conserved during tunneling. They make a reference to a theory paper. However, since this is a crucial assumption of the model, it would be nice to support that claim by physical arguments and eventually data. I therefore strongly encourage the authors to add further information in the SM to sustain that claim.

- In Fig. 1h, the authors evaluate T_K . First, as the authors know T_K is a crossover scale and therefore there are many possible definitions of T_K . I guess they used the NRG one. However, it would be better to clarify that point. Second, it seems the 1h case (yellow arrow) has a T_K such that $T_{\text{exp}}/T_K \sim 0.3$. A priori, this is far from the Fermi liquid regime where the impurity is fully screened (the singlet is not yet fully formed). In other words the Kondo entanglement which governs the main results of that paper is a priori not fully operative. On the theory side, the authors may add some analysis in the SM that despite T is not one order of magnitude smaller than T_K , the singlet is already almost formed. This can be done by calculating the residual entropy in DM-NRG or directly the spin-spin correlation function characterizing the singlet state.

This is an important element to support the whole consistence of the interpretation.

- The authors choose to plot d^2/dV^2 while most papers plot dI/dV . Why such choice ?

I also have minor comments and questions:

- Fig. 1e is merely not visible and does not bring much to the analysis. Maybe the authors could add a zoom on one peak in some inset. Furthermore, they mention that the width and height are underestimated. I got the visual impression that the theory overestimates the height of the peak. Comments ?

- The authors have a side-remark: 'Consequently, a Kondo revival is also observed in the 1h valley at fields of about 1.2 Tesla'.

First, I do not understand the consequently. Second it would be nice to better characterize such a

revival (make a 2D cut of dI/dV for example) despite this is not the main focus of the paper.

- I would encourage to put details supporting Eq. (1) in the Methods section since this is the unique equation of the paper. Just move some of the SM in the methods.

- I did not understand clearly the meaning of the notation page 3 of $J_u^2=J_d^2=0$. Please clarify/define these notations.

- On the same footing, the authors should define their notation concerning the entanglement configuration more properly.

- In the caption of Fig. 3, I guess the authors should change spin into pseudo-spin $\{\text{cal I}\}$.

To summarize, this paper demonstrates some new non-equilibrium features associated with the formation of the Kondo singlet.

The overall analysis is convincing. This careful analysis thus adds some new side-features on the already highly studied field of Kondo effect in quantum dot.

I definitely think that this paper must be published in some form provided the authors take into account my aforementioned comments.

Reviewer #2 (Remarks to the Author)

The authors present a comprehensive experimental and theoretical treatment of a clean carbon nanotube quantum dot in the Kondo regime. Despite significant effort to understand fully the Kondo effect in carbon nanotubes including both spin and pseudospin degrees of freedom over more than 10 years, questions remained about missing transport resonances, which this work finally clarifies. Based on the quality of both the experimental data and analysis, I believe this manuscript is suitable for publication in Nature Communications after the questions below are addressed satisfactorily.

1. An experimental temperature of 30 mK is stated. Is this a (measured) electron temperature or a mixing chamber temperature? The supplement lists a temperature parameter for theory of 232 mK. What is the origin of this discrepancy? Is the actual temperature of the electrons in the experiment really closer to 200 mK? It appears that the electron temperature should be measurable on the electron side of the gate voltage range where the device appears to fully pinch off (Fig. S4), though without absolute electron numbers noted, whether this will be possible is difficult to tell from the data provided.

2. Related to the above, it is noted that the experimental peaks are broader than the nominal temperature. The way this sentence is phrased, it sounds as if it should be expected that the peaks are broadened by temperature rather than tunnel coupling, when the latter will always be the case for a Kondo dot in this regime. What's surprising is that although the transport calculations are "exact up to second order in the tunnel coupling", they apparently fail to capture the most basic, non-Kondo term that is first-order in tunnel coupling. Please provide some comment about this particular failure of the model.

3. Within a quartet of charge states, it appears that a single Kondo temperature is used. In a single-gate device, however, it is not possible both to keep tunnel couplings constant and change the charge state. Is there a way to see that T_K is approximately constant in the range of gate voltages required to produce Fig. 1d, for example?

4. In Fig. 1e and f, theory also does not accurately reproduce the positions of the peaks, particularly for the 3e-4e transitions and 1h-2h transitions. What is the origin of this discrepancy, which is not small, accounting for approximately 5-10% of the value of U ?

5. In the last paragraph of page 3, it is stated that "the localized CNT pseudospin is fully screened by an opposite net pseudospin in the leads". Are the leads here considered to be CNT or Pt? If the latter, this sentence is confusing since a pseudospin degree of freedom does not exist to my knowledge in Pt. Is the idea that an electron from the lead first tunnels from the Pt to a segment of CNT outside the dot?

6. Related to 5. above, it would be nice to include a picture of the actual device, at least in the supplement, to help understand the geometry. The geometry is by now a relatively standard one, but seeing the precise dimensions can help clarify questions about what exactly the experimental situation is.

7. In Fig. 4d, the experimental data at positive bias changes color from blue to red above the C transition. The theoretical plot has a monotonically decaying blue color above C. In the experimental data, a zero of d^2I/dV^2 indicates a transition at that location, which is where the P transition would be. Is the P transition weakly present in this case, and if so, why?

8. Overall the manuscript is extremely well-written, clear, and easy to follow. However, I believe one element that is lacking is a clear picture, in language that more experimentalists could appreciate, for why the P resonances get blocked. The theoretical details are described on page 3, but I believe as written, it is difficult to come away with an appreciation for why this effect (that is the main observation of the manuscript) occurs.

Reviewer #3 (Remarks to the Author)

The manuscript by Niklas et al analyses Kondo features observed in a carbon nanotube quantum dot. In particular it focuses on a set of inelastic cotunneling lines observed for weak coupling of the quantum dot to the leads (very small Kondo temperature) but not for strong coupling (large Kondo temperature). The authors show that Kondo correlations lead to screening of the Kramers pseudospin of the dot by the conduction electrons; a consequence of which is that (at finite bias) only the so-called T and C transitions are expected in this regime - in good agreement with experimental data. This is in contrast to the weak coupling regime where P transitions are also expected and observed. Here both regimes are studied in the same device.

I find this work impressive on a technical level (both experiment and theory/analysis). However, it is not that clear to me how it substantially differs from that of Ref. [12], which has several authors in common with the current manuscript of Niklas et al: the conclusions of these two papers are essentially the same. Both regimes (weak and strong coupling) have been studied before but the authors argue that their manuscript stands out because both regimes are studied here in the same device. However, I do not think that this resolves any outstanding questions; i.e. I do not think that previous conclusions (e.g. those of Ref. [12]) were in doubt.

I have several minor comments:

- The manuscript mentions an experimental temperature " $T_{\text{exp}} = 30 \text{ mK}$ ". Is this the electron temperature or lattice temperature? How is T_{exp} determined for this data set?

- The manuscript describes Kondo temperatures as either, very small, or large or of $0.1 < T_{\text{exp}}/T_K < 1$; but I couldn't find numbers for T_K in the manuscript or how the T_K are obtained from the data.

- The final paragraph is not very precise and speculative (this in contrast to the rest of the paper). For example, what is meant by 'dark side of the Kondo effect'? I understand that the authors would like to provide an outlook but the link with quantum information processing seems somewhat artificial. It is not clear to me how quantum information would be stored in the Kondo

singlet or the order of magnitude of any coherence times.

In conclusion, while the manuscript is sound I believe it to be suitable for a more specialized journal.

Reply to Reviewer 1

We thank the referee for his/her very careful and constructive review of our manuscript. We are pleased that he/she found “the data very clean”, “the paper very nicely and clearly written”, and that “the agreement theory/experiment is quite remarkable”. Furthermore he/she “definitely” thinks that “the paper must be published in some form” provided that his/her comments are accounted for. We have indeed fully accounted for the referee’s criticisms in the revised version of the manuscript, as detailed in the following.

1. The reviewer asks what is the origin of the discrepancies between the DM-NRG calculations and the data in the comparison shown in Fig. 1f, and why we should use different parameters for the NRG approach compared to other methods.

Answer: We believe that the discrepancy is due to the fact that in the DM-NRG calculations no exchange coupling J was included. Together with spin-orbit coupling and valley mixing it contributes to the $SU(4)$ symmetry breaking in the valley with $N=2$ (see e.g the spectrum in Fig. 2b), and hence the experimental conductance is more rapidly suppressed in that valley than as predicted by our simulations. More specifically, the symmetry is $SU(2) \times SU(2)$ when $J=0$. A finite J together with spin-orbit coupling and valley mixing also breaks the $SU(2)$ symmetries in valley $N=2$. On the other hand, J is not relevant for describing the spectrum for the $N=1$ and $N=3$ cases (Figs. 2a, 2c), which was in the focus of the present work.

The inclusion of J in the DM-NRG calculation is possible but at the expense of a significant increase in computational cost due to the reduced symmetry and hence we have preferred not to include it. Likewise J was not included in the KEA calculations.

We have added the explanation above on page 2 (top of second column) to better clarify this point.

2. The reviewer "strongly encourages" to add further information in the supplementary material (SM) to support the claim that the CNT quantum numbers are conserved during tunneling.

Answer: We have added a paragraph together with a new figure showing the experimental device in Sec. I of the SM to support our claim. The figure is the new figure S1.

3. The referee asks to clarify which definition of the Kondo temperature T_K is used in Fig. 1h. Moreover, he/she observes that because the experiment is in the cross-over regime (characterized by ratios of the experimental temperature T_{exp} and the Kondo temperature T_K in the range $0.1 < T_{\text{exp}}/T_K < 1$), the system is "far from the Fermi liquid regime where the impurity is fully screened." He/she thus suggests to add in the SM some analysis to show that despite T_{exp}/T_K is not one order of magnitude smaller than T_K "the singlet is already almost formed". To this aim he suggests to calculate the residual entropy or directly the spin-spin correlation function.

Answer: We have added the definition of T_K in the caption of Fig. 1.

Regarding the formation of the Kondo singlet at the temperature of the experiment, we have calculated the DM-NRG impurity entropy as suggested by the referee. The results are reported in the new section II of the SM with the impurity entropy and specific heat being shown in the new Fig. S2 as a function of temperature. They yield a regime where the entropy is close to zero up to $T \sim 0.01T_K$, (i.e the system is in the groundstate singlet), and confirm the persistence of Kondo universality up to temperatures $T \sim T_K$.

4. The referee asks why we choose to plot the current second derivative d^2I/dV^2 rather than the dI/dV , as done in "most papers".

Answer: We decided to plot d^2I/dV^2 in the main text to enhance eye visibility of the excitation spectra. For completeness, the dI/dV stability diagrams are reported in Figs. S4 and S5 of the SM.

A similar representation of the data in terms of the current second derivative was already used in Refs. 16 and 17.

We added a comment on page 2 (second column) to clarify this point.

5. The referee also has minor remarks and questions.

- He/she says that "Fig. 1e is merely not visible and does not bring much to the analysis."
He/she asks to add one zoom in Fig. 1e to better resolve the Coulomb peaks and to better comment on the comparison between theory and experiment

Action: We have modified Fig. 1e by separating the experimental from the theoretical data and rephrased the paragraph on page 2 discussing the comparison. We chose to do this rather than to include a zoom of one of the peaks because there are too few experimental data points in the vicinity of the peaks.

- He/she is confused by the adverb "consequently" used in the sentence on page 2 "Consequently, a Kondo revival is observed in the 1h valley at fields of about 1.2 Tesla." Moreover, he/she asks to "better characterize" the Kondo revival by making for example a 2D cut of the dI/dV .

Action: We have rephrased the sentence (bottom of page 2). Moreover we refer to Fig. S3 of the supplement where bias traces of the dI/dV showing the revival are reported.

- He/she encourages us to put details supporting Eq. (1) in the Methods section "by moving some of the SM in the methods."

Action: We have followed the suggestion of the referee and moved some material of the SM to the Methods section

- He/she asks to clarify the notation $J_u^2=J_d^2=0$ on page 3

Action: We have clarified the notation

- He/she asks to define the notation concerning the entanglement configuration more properly

Action: We have improved our notation concerning the entanglement configuration

- He/she suggests "to change spin into pseudo-spin" in the caption of Fig. 3

Action: We have done the change

Reply to Reviewer 2

We thank the referee for his/her careful reading of the manuscript and the constructive comments. We are grateful to the referee for acknowledging that "despite significant effort to understand fully the Kondo effect in carbon nanotubes ... over more than 10 years, questions remained about missing transport resonances, which this work finally clarifies." He/she recommends publication in Nature Communications given that his/her questions are "addressed satisfactorily".

A detailed answer to the reviewer's questions is given below. We have fully accounted for his/her remarks which, we believe, have improved the readability of the manuscript in a way that a broader audience can appreciate. We thank the referee for stressing this point.

1. The referee asks if the experimental temperature of 30 mK is a measured electron temperature or a mixing chamber temperature. He then asks why in the supplement a theory temperature

of 232 mK is listed and if the actual temperature in the experiment is closer to 200 mK. Finally, he asks about the possibility of measuring the electron temperature on the electron side of the gate voltage.

Answer: The temperature of 30 mK is a mixing chamber temperature. Hence the actual temperature might be larger. As explained in the answer to question 2, it was not possible to extract the actual temperature of the experiment, not even from the electron side. Thus the mixing chamber temperature of $T=30$ mK was used for the DM-NRG simulations. For the perturbative calculations (PT) and the nonperturbative Keldysh effective action (KEA) calculations in Figs. 2, 3 and 4 a temperature of 232 mK and of 0K, respectively, was used.

The information regarding the temperature in the PT, NRG and KEA calculations has been added to Sec. III of the supplement.

Why a temperature of 232 mK has been used in the PT calculations is explained in the answer to question 2 below.

2. The referee asks to clarify the issue related to the fact that the experimental peaks are broader than the nominal temperature while our perturbation theory (PT) does not capture this feature.

Answer: The validity of our perturbation theory requires small ratios Γ/U and Γ/T , where Γ is the tunneling coupling and U the charging energy. I.e., Γ is the smallest energy scale in the problem. In this regime the temperature smearing of the Fermi function dominates over the tunneling-induced broadening. Correspondingly, the PT predicts that the width of the resonance features is proportional to the temperature. This is the case for the theoretical Coulomb oscillation peaks shown in Fig. 1e, which already are captured by “the most basic, non-Kondo term that is first-order in tunnel coupling”, as pointed out by the referee. Likewise for the cotunneling resonances in the theoretical parts of Figs. 2d-f and 2h, which originate from second-order virtual processes.

When $\Gamma > T >> T_K$ (and still $\Gamma/U \ll 1$) tunneling induced broadening dominates over thermal smearing. In this so called intermediate coupling regime Kondo correlations are not important but Coulomb oscillations and cotunneling resonances become tunneling broadened. This is what is seen in the experimental data shown in Fig. 1e and Figs. 2d-f, 2h. This feature can be predicted only by nonperturbative transport theories, where broadening originates from virtual tunneling processes to all order in Γ (see e.g. Refs. 31- 33). Notice that this also implies that the actual temperature of the experiment cannot be extracted from an analysis of the data in the electron regime.

Crucially, both in the perturbative regime as well as in the intermediate coupling regime the cotunneling transport spectrum, reported in Figs. 2d-f, 2h, is directly related to the excitation spectrum of the isolated quantum dot. Because this was the main information we wanted to extract from the data, we have used the simpler PT (with all tunneling processes up to second order in Γ being retained).

By this choice, the ratio Γ/T is constrained to be smaller than one and cannot be fitted to the experiment; it becomes a free parameter (notice the a.u. in the PT plots). Having thus some

freedom, we have chosen the temperature of 232 mK to get a larger broadening of the PT resonances.

In order to better explain that the experiment in the electron side of the gate voltage is already in the intermediate coupling regime, we have modified the paragraph on page 2 discussing Fig. 1e.

3. The referee asks to clarify whether a single Kondo temperature was used within a quartet of charge states, and if T_K is approximately constant in the range of gate voltages required to produce Fig. 1d.

Answer: In general, T_K is a nontrivial function of the charging energy U , the value ϵ_i of the i^{th} dot energy level with respect to the Fermi level of the leads, and the couplings Γ_i . In the DM-NRG calculation of Fig. 1f we kept both the couplings Γ_i and the energy U constant within the quadruplet. However, because in our set-up the electrochemical energies ϵ_i account for the effects of the back-gate voltage, their value varies as the gate voltage is changed. As a consequence, also T_K varies as the gate voltage is changed from the vicinity of a Coulomb peak to the middle of a valley, or from valley to valley. For example, our DM-NRG calculation in Fig. 1h predicts a Kondo temperature in the middle of the 3h valley being approximately twice the one in the middle of the 1h valley.

The estimated values of T_K in the valleys 1h and 3h are now explicitly given on page 2 (top of second column).

4. The referee remarks that "In Fig. 1e and f, theory also does not accurately reproduce the positions of the peaks, particularly for the 3e-4e transitions and 1h-2h transitions". He/she asks for the origin of this discrepancy.

The referee correctly observes some discrepancy in the theoretical and experimental peak positions both for the electron and the hole regimes. It has a different origin for the two regimes. Electron regime: At positive gate voltage the system is in the intermediate coupling regime $\Gamma > T$, where virtual charge fluctuation processes give rise to broadening (as explained at point 2) but also to Lamb-shifts (see e.g. Ref. 33). Such shifts affect the peaks positions and are not included in the perturbative calculation. Hence, fixing U , the exchange J and the Kramers energy splitting Δ is not sufficient to get an accurate fit of the whole quadruplet of peaks.

Hole regime: In the DM-NRG charge fluctuations are naturally accounted for. We believe that the discrepancy is due to the fact that in the DM-NRG calculations no exchange coupling J was included. The latter contributes to the $SU(4)$ symmetry breaking in the valley with $N=2$ (see e.g. the spectrum in Fig. 2b), and hence the experimental conductance is more rapidly suppressed in that valley than as predicted by our simulations. On the other hand, J is not relevant for describing the spectrum for the $N=1$ and $N=3$ cases (Figs. 2a, 2c), which was the focus of the present work. The inclusion of J in the DM NRG calculation is possible but at the expense of a significant increase in computational cost. This is because when accounting for J , spin-orbit coupling and valley mixing also the $SU(2) \times SU(2)$ symmetry is broken in valley two. Because the quantitative behavior in valley two was not in the main focus of our work, we have not included J .

We have added a sentence on page 2 (top of second column) clarifying this point.

5. The referee asks to better clarify the sentence on page 3 stating “the localized CNT pseudospin is fully screened by an opposite net pseudospin in the leads.” He asks whether “the idea is that an electron from the lead first tunnels from the Pt to a segment of CNT outside the dot?”

Answer: Yes, we assume that an electron from the leads first tunnel to a CNT segment outside the dot. We have added a clarifying sentence on page 3.

6. Related to question 5, the referee suggests “to include a picture of the actual device, at least in the supplement, to help understanding the geometry.”

Answer: We thank the referee for the suggestion. We have added a figure and a sketch of a device similar to the one used in the experiment in the supplement (see new Fig. S1).

7. The referee notices that “In Fig. 4d, the experimental data at positive bias changes color from blue to red above the C transition.” He asks whether this indicates that the P transition is “weakly present” at that location.

Answer: A maximum of dI/dV corresponds to a zero of the d^2I/dV^2 . Near the resonance the d^2I/dV^2 decreases monotonically and hence the data change color from red to blue upon increasing the bias. A change from blue to red in the d^2I/dV^2 indicates a minimum in the differential conductance. Thus the feature above the C transition does not seem to be directly related to the P transition.

We have better explained the meaning of the color scale by adding a sentence in the caption of Fig. 2.

8. The referee states that “the manuscript is extremely well-written, clear and easy to follow. However, I believe one element that is lacking is a clear picture, in language that more experimentalists could appreciate, for why the P resonances get blocked.”

Answer: We thank the referee for this constructive remark. We have put extra effort to clarify why the P resonances get blocked. In particular, on page 4 we clarify that while the C and T transitions connect the singlet groundstate to excited singlets; the P transition seem to connect the singlet groundstate to an excited triplet state.

Reply to Reviewer 3

We thank the referee for his/her positive comments on our manuscript: “I find this work impressive on a technical level (both experiment and theory/analysis).” However he/she criticizes that “it is not clear to me how it substantially differs from that of Ref. [12]” since “the conclusions

of these two papers are essentially the same.” Moreover he/she does not think that having studied the weak and strong coupling regime in the same device “resolves any outstanding questions; i.e., I do not think that previous conclusions (e.g. those in Ref. [12]) were in doubt”. As such the reviewer believes the manuscript to be suitable for a more specialized journal.

Besides this main criticism the referee has some minor comments.

A detailed answer to the reviewer’s main criticism as well as to the minor ones is given below.

First of all we share the reviewer’s opinion that the previous conclusions, in particular those in Ref. [12], were not in doubt. However, as we argue below, we think that the present manuscript represents a substantial progress with respect to Ref. [12], both in the understanding of the blockade mechanism in the Kondo regime as well as in its manifestation.

We identify three important points:

- i) We believe that having shown the presence of the P transitions in the cotunneling regime and their absence in the Kondo regime on the same device is important per se as a remarkable manifestation of emerging Kondo screening.
- ii) New with respect to Ref. [12], and in our opinion a most substantial advance, is the **identification of the observables being screened**. We have demonstrated that these are the Kramers charge and pseudospin associated to the each Kramers doublet. Eq. (1) is the key to understand this statement: by rewriting the single-particle CNT Hamiltonian in terms of Kramers charges and pseudospins, we could associate proper pseudospin quantum numbers to each spectral line observed in magnetic field, independent of the field amplitude or of the polar angle θ formed by the magnetic field and the CNT axis, as shown in Figs. 4a and 4b. We are convinced that the results shown in Figs. 4c and 4d are not only “impressive on a technical level”, rather they beautifully demonstrate for the first time that the proper degrees of freedom to be considered are neither the true spin or orbital degrees of freedom of the CNT but the more abstract pseudospins.
- iii) A related aspect regards the **many-body excitation spectrum** in the Kondo regime. It is well-known that the Kondo ground state is a singlet, but the nature of the low lying excitations states of a $SU(2)\times SU(2)$ Kondo system has not been much investigated so far. Our theoretical and experimental analysis indicates that for $SU(2)\times SU(2)$ systems also the excited doublet of the impurity gives rise in the Kondo regime to an excited singlet with total zero pseudospin. Correlations yielding these singlet states seem to be robust, in the sense that they persist also at the finite temperatures of the experiment where, as seen in figure 1h and in the new figure S2 of the supplement, the system is in the cross-over regime for both the 1h and 3h valley.

We have better emphasized points ii) and iii) in the revised version of the manuscript by rewriting some paragraphs on page four, where also conclusions are drawn.

The reviewer had minor comments:

- He/she asks whether T_{exp} is “the electron or lattice temperature” and how it is determined for this data set

Answer: T_{exp} is the mixing chamber temperature. We have clarified this point on page 1.

- He/she asks to provide the numbers for the Kondo temperatures and how they are obtained from the data

Answer: We apologize for not having written down explicitly the values of T_K estimated for our experiment.

In the revised version we have added on top of page 2 the values of T_K close to the middle of the 1h and 3h valley as provided by our DM-NRG calculations. The Kondo temperature is obtained from the linear conductance $G(T, V=0)$ by requiring that $G(T_K, V=0)=0.5 G(T=0, V=0)$.

The experiments have been performed at fixed temperature while varying the bias and gate voltages. Thus what is available from the experiment are bias traces (at fixed gate voltage) providing the differential conductance $G(T, V)$ as shown in Fig. S3(c). To obtain T_K from the experimental bias trace data requires to know the relation between $G(T, V)$ evaluated at a reference voltage V^* and $G(T, 0)$. Such relation is known for the SU(2) Kondo effect (where $G(T=0, V_K) \sim 0.67 G(T=0, 0)$ with $eV_K = k_B T_K$, see e.g. Kretinin et al., PRB 85, 201301 (2012); Pletyukhov and Schoeller, PRL 108, 260601 (2012)), but not for the more complex SU(2)xSU(2) case investigated here. As explained in Sec. III of the supplement, to qualitatively compare theory and experiment we have scaled the respective theoretical and experimental curves by a voltage V^* such that $G(T=0, V=V^*)=0.8 G(T=0, 0)$ for the KEA and $G(T_{\text{exp}}, V=V^*)=0.8 G(T_{\text{exp}}, 0)$ for the experiment.

- The referee asks to better explain the ideas behind the final paragraph of the manuscript which he/she finds “not very precise and speculative”. He asks what is meant by “dark side of the Kondo effect” and “how quantum information would be stored in the Kondo singlet or the order of magnitude of any coherences times.”

Answer: By “dark side of the Kondo effect” is meant that Kondo correlations not only open new transport channels (not accessible in the weak coupling regime due to Coulomb blockade), but also block some of the channels accessible at weak coupling. We understand that the word “dark” might be misleading and we removed it. Moreover, we have rewritten that conclusion paragraphs. We now discuss the occupation of the groundstate singlet and the robustness of Kondo correlations against thermal fluctuations, as they results from a DM-NRG study of the impurity entropy reported in the new Sec. II of the Supplement.

Main changes to the manuscript

- i) We have improved the text in several parts according to the suggestions of the referees
- ii) We have modified slightly Figs. 1 and 4
- iii) We have moved the part regarding the $SU(2)\times SU(2)$ representation of the CNT model Hamiltonian from the Supplement to the Methods, as suggested by reviewer one
- iv) We have added the new figure S1 in the supplement showing an experimental set-up similar to the one used in the actual experiment, as suggested by reviewers one and two
- v) We have added the new section II to the supplement discussing the impurity entropy and specific heat, together with the new figure S2, as suggested by reviewer one.

In summary, we believe that our work reports substantial novel physics, of interest for the broad readership of Nature Communications. Our clean experiments supported by a comprehensive theoretical analysis shed new light on complex, strongly interacting nonequilibrium nanosystems, with applications to various fields of active fundamental research.

Reviewers' Comments:

Reviewer #1 (Remarks to the Author)

I am satisfied by the modifications made by the authors following the various constructive remarks and comments made by all referees. The manuscript together with the new material provided in the SM strengthen the present manuscript.

Reviewer #2 (Remarks to the Author)

[The referee recommends publication of the paper with no further comments for the authors]